# Palladium(II)-Catalyzed Efficient Synthesis of Wedelolactone and Evaluation as Potential Tyrosinase Inhibitor

**DOI:** 10.3390/molecules24224130

**Published:** 2019-11-15

**Authors:** Huidan Huang, Jianqiu Chen, Jie Ren, Chaofeng Zhang, Fei Ji

**Affiliations:** 1College of Engineering, China Pharmaceutical University, #639 Longmian Avenue, Jiangning District, Nanjing 211198, China; zgykdxbjl@163.com; 2State Key Laboratory of Natural Medicines, Research Department of Pharmacognosy, China Pharmaceutical University, #639 Longmian Avenue, Jiangning District, Nanjing 211198, China; cpubjl@163.com

**Keywords:** wedelolactone (WEL), tyrosinase, Suzuki–Miyaura reaction, inhibitors

## Abstract

Tyrosinase is an enzyme widely distributed in nature, which has multiple functions, especially in the melanin biosynthesis pathway. Despite the few clinically available tyrosinase inhibitors for whitening, a great demand remains for novel compounds with low side effects in terms of potential carcinogenicity and improved clinical efficacy. A natural product, wedelolactone (WEL), with a polyhydroxyl moiety, attracted our attention as a potential tyrosinase inhibitor. Before we studied the biological activity of the natural product, a synthetic methodological research was firstly carried to obtain enough raw material. WEL could be obtained efficiently through palladium-catalyzed boronation/coupling reactions and 2,3-dicyano-5,6-dichlorobenzoquinone (DDQ)-involved oxidative deprotection/annulation reactions. Immediately after, the natural product was proven to be an efficient tyrosinase inhibitor. In conclusion, we developed a mild and efficient approach for the preparation of WEL, and the natural product was disclosed to have anti-tyrosinase activity, which could be widely used in multiple fields.

## 1. Introduction

Tyrosinase (EC 1.14.18.1) is an enzyme widely distributed in nature, which has multiple functions, especially in the melanin biosynthesis pathway [1]. The enzyme can catalyze l-tyrosine to l-3,4-dihydroxyphenylalanine (l-DOPA), with further oxidation to dopaquinone [2]. Dopaquinone transforms from brown to black through several reactions. Abnormal melanin production, including melasma, freckles, lentigo, senilis, and other forms of melanin hyperpigmentation could be a serious aesthetic problem [2,3]. In addition, tyrosinase is also involved in the defensive and developmental functions of pests [4]. Excessive dopaquinone was also reported to cause neurodegeneration related to Parkinson’s disease [5]. Thus, many tyrosinase inhibitors are applied in cosmetics and pharmaceutical products [6]. Despite the few clinically available tyrosinase inhibitors for whitening, a great demand remains for novel compounds with low side effects in terms of potential carcinogenicity and improved clinical efficacy [7,8]. Obviously, more efforts are still needed in that direction; therefore, we recently focused our interest on discovering novel tyrosinase inhibitors. 

Compounds with the polyhydroxyl moiety were proven to be potential tyrosinase inhibitors [9,10,11]. Recently, our group undertook research on a natural product with a polyhydroxyl moiety named wedelolactone (WEL), which is derived from the medical plant *Eclipta prostrata* [12]. Although a wide range of pharmacological activities of WEL were reported, there is less information on the inhibitory effect and reversibility of WEL on tyrosinase. Thus, the inhibitory activity and mechanism of WEL toward tyrosinase deserves deeper investigation; however, but the present knowledge on synthesis of the natural product is limited. Although several groups invested substantial effort in the preparation of WEL, these methods had several disadvantages, including a time-consuming nature with complicated synthetic approaches [13,14,15]. 

Among these methods, two routes listed in Figure 1 are commonly recognized by the industry. However, both methods have several disadvantages. The first method (reported by Yang [14]) involves a crucial intermediate, phenyl acetylene, which is difficult to prepare. The route has a low 15% overall yield with a long linear sequence (total of 12 steps), and it is rarely applied to access a variety of WEL analogues for structure transformations. The second method (reported by Lee et al. [13]) employs toxic organotin and organomercurial reagents, which limit industrial production and increase operation complexity. In addition, both methods can only obtain the natural products on a small scale. As the present methods are imperfect and unsatisfactory for further investigation of WEL as an efficient tyrosinase inhibitor, the development of a facile, versatile, and mild approach is urgently needed. 

## 2. Results and Discussion

### 2.1. Palladium(II)-Catalyzed Efficient Synthesis of WEL

Retrosynthetically, WEL could be logically disconnected by the ring opening of furan to afford the intermediate **4**, which is further disconnected by C–C bond cleavage to trace back to the intermediate 3-bromo-5-benzyloxy-7-acetoxyl-2-chromenone **3** and the readily prepared 4,5-dibenzyloxy-2-(4-methoxybenzyl)oxy-phenyl boronic ester **2** (Scheme 1). This similar synthetic strategy was ever used by Shen for the synthesis of hirtellanine A [16]. Synthetically, we expected that polysubstituted coumarin **4** could be obtained by Pd(II)-catalyzed Suzuki–Miyaura coupling of 3-bromocoumarin **3** and polysubstituted phenyl boronate ester **2** which could be generated by a Pd(II)-catalyzed boronation reaction of the polysubstituted bromobenzene **1**. The coupling product **4** then underwent a DDQ-oxidation deprotection/annulation reaction to deliver the final product WEL **5**.

In the beginning of our synthesis, we focused on the generation of the polysubstituted bromobenzene **1** (Scheme 1). Selective protection of the three phenolic hydroxyl groups presented a big synthetic challenge. After reviewing the literature [16,17], we chose the commercially available 3,4-dihydroxybenzaldehyde **6** as the starting material to provide the polysubstituted bromobenzene **1** via the *m*CPBA-mediated Baeyer–Villiger oxidation strategy, which resolved the selective protection of phenolic groups. Synthetically, protection of the phenol groups of 3,4-dihydroxy-benzaldehyde **6** with benzyl bromide afforded 3,4-bis(benzyloxy)benzaldehyde **7** in 86% yield, which subsequently underwent *m*CPBA-mediated oxidation and hydrolysis to deliver 3,4-bis(benzyloxy)phenol **8** in 87% yield. Next, **8** was firstly protected with PMBCl and then selectively brominated with NBS to obtain polysubstituted bromobenzene **1** in high yield.

Subsequently, we aimed to synthesize 3-bromo-5-benzyloxy-7-acetoxyl-2-chromenone **3** (Scheme 2). According to reported methods [14], the commercially available phloroglucinol **10** and ethyl propiolate as starting materials smoothly underwent ZnCl_2_-catalyzed esterification and cyclization reactions to provide dihydroxycoumarin **11**. Dihydroxycoumarin **11** was then treated with acetyl chloride to deliver the corresponding diacetoxylcoumarin **12** in good yield. The following reaction involved the dibromination and dehydrobromination of ketene **12** to afford 3-bromo-5,7-diacetoxyl-2-chromenone **13** in 75% yield. Reaction temperature is vital to the bromination reaction. It was found that maintaining the reaction temperature at 10 °C could avoid the bromination of acetyl group of **12** and acquire the optimal reaction yield. Finally, we ran the partial benzylation of 3-bromo-5,7-diacetoxyl-2-chromenone **13**. The similar reactivity of the 5- and 7-hydroxy groups did not facilitate the selective 5-benzylation of **13**. According to Humbert’s selective alkylation reaction conditions [18], we tried similar normal benzylation conditions (BnBr, K_2_CO_3_, acetone, reflux) and found that traces of water in acetone chemoselectively hydrolyzed the 5-acetyl group as well, generating the intermediates **14** and **15**, stabilized by the conjugating effect of the pyrone ring. Although the undesired deacetylation byproducts still existed, the major benzylating product was 3-bromo-5-benzyloxy-7-acetoxyl-2-chromenone **3**. With the use of ethanol recrystallization, the crude product **3** was obtained in 65% yield and directly used without further purification for the next step.

With the key intermediates polysubstituted bromobenzene **1** and 3-bromo-5-benzyloxy-7-acetoxyl-2-chromenone **3** in hand, we focused on the combination of two fragments via boronation and a subsequent coupling reaction (Scheme 3). Under strong alkaline and low-temperature conditions [19], polysubstituted bromobenzene **1** could be treated with bis(pinacolato)diboron to afford the borate ester **2**. To simplify the operation procedure, Pd(II)-catalyzed boronation was attempted, and the corresponding borate ester **2** was obtained in quantitative yield as well. The formed crude product **2** could be utilized directly without further purification for the following Suzuki–Miyaura reaction and afforded the deacetyl coupling product **16** in good yield. To avoid the stability issue of borate ester in column chromatography, the boronation and subsequent coupling reaction were carried out conveniently in one pot and gave a 72% overall yield. The one-pot reaction conditions were optimized by initiating an experiment to evaluate the effect of the various parameters on the reaction yield (see Appendix A). Methylation of the deacetyl coupling product **16** in the standard treatment with methyl iodide gave polysubstituted coumarin **4** in 85% yield.

Finally, the completion of the remaining steps in WEL synthesis required PMB deprotection, cyclization, and debenzylation. Removing the PMB on the substituted coumarin **4** in acidic conditions (HOAc, reflux) led to the decomposed mixture and gave an unsatisfactory result [20]. It was reported that DDQ could also be used as a deprotecting reagent to remove the PMB group [21]. Thus, we adopted this synthetic strategy and hoped that oxidation cyclization could subsequently happen after the deprotection. Fortunately, DDQ-involved deprotection and oxidative annulation proceeded well as anticipated, forming the desired product 1,8,9-tris(benzyloxy)-3-methoxy-6*H*-benzofuro[3,2-c]chromen-6-one **17** in 56% yield (Scheme 4). Compound **17** was then treated with BCl_3_ as a debenzylating reagent, generating the final product wedelolactone **5** in 81% yield. ^1^H- and ^13^C-NMR spectra of the synthetic product were in agreement with reported data for WEL [13].

### 2.2. Biological Activity

With a sufficient amount of WEL in hand, we tested the in vitro tyrosinase activity. Kojic acid was used as a positive control, as typically employed in the evaluation of tyrosinase inhibitors. As shown in Figure 2, WEL caused strong tyrosinase inhibition in a concentration-dependent manner. The 50% inhibitory concentration (IC_50_) values of WEL and kojic acid were determined to be 1.2 ± 0.3 and 14.2 ± 1.6 μM, respectively. The strong tyrosinase inhibitory activity may be due to the multiple hydroxyl groups in the structure. However, this needs to be further confirmed by determining the derivatives of WEL (data not shown). To confirm the inhibitory mechanism of WEL against tyrosinase, the plots of initial velocity versus tyrosinase concentration at different concentrations of WEL were developed, and a set of straight lines was obtained (as shown in Figure 2). All of the lines passed through the origin, and an increase in the WEL concentration reduced the slopes of the lines, indicating that the compound was a reversible inhibitor. 

### 2.3. The Docking Studies

Computational docking studies were employed to determine the preferred binding sites of WEL in tyrosinase using the GOLD5.1 software. Tyrosinase contains two copper ions, and each copper ion is coordinated by three histidine residues. The first copper (Cu A) is coordinated by His61, His85, and His94, and the ligands of the second copper ion (Cu B) are His259, His163, and His 296 [21]. WEL could insert into the active site with a copper domain and it was found to interact with various amino-acid residues (Figure 3). The possible site of hydrogen-bonding interactions of WEL with tyrosinase was Asn260. The hydrogen-bonding residues could affect the binding affinity considerably. According to the molecular docking study, it was found that the His61, Val248, His259, Asn260, and His263 amino-acid residues of tyrosinase interact with WEL. The enzyme kinetic analysis and molecular docking studies confirmed that WEL binds to tyrosinase in the active site.

## 3. Materials and Methods 

### 3.1. Reagents and Materials

LC-2010 HPLC system (Shimadzu) equipped with a Phenomenex Luna C18 analytical column (250 mm × 4.6 mm, 5 µm), a 6520 Accurate-Mass Q-TOF LC/MS system (Agilent, Santa Clara, CA, USA) were obtained for this study. The chemical reagents, TYR and L-tyrosine were purachased from m Sigma-Aldrich (St. Louis, MO, USA). Koic acid was purchased from Shanghai Yuanye biological technology Co., Ltd. (purity >98%; Shanghai, China). Ultrapure water was prepared using a Millipore water purification system (Millipore, Bedford, MA, USA).

### 3.2. Chemical Synthesis

Preparation of *3,4-bis(benzyloxy)benzaldehyde*
**7**: A 250-mL reaction vessel with a magnetic stirring bar was equipped with 3,4-dihydroxybenzaldehyde (5.13 g, 37.2 mmol), K_2_CO_3_ (25.70 g, 186.0 mmol), BnBr (19.0 g, 111.6 mmol), and DMF (100 mL). The mixture was stirred at room temperature for 4 h. After the reaction was completed, water (500 mL) was added, and the mixture was extracted with ethyl acetate (500 mL). The organic layer was washed with water (250 mL × 2) and brine (250 mL). After drying over MgSO_4_, the extracts were filtered and concentrated. The residue was firstly stirred in hexane (200 mL) overnight and then filtered to afford the intermediate **7** without any purification for the next step. White solid; 10.17 g, 86% yield; melting point (m.p.) 89–90 °C; R_f_: 0.43 (EA:hexane = 1:4); IR (cm^−1^): 3026, 2819, 2726, 1676, 1596, 1580, 1512, 1435, 1396, 1386, 1349, 1269, 1245, 1231, 1211, 1135, 1021; ^1^H-NMR (CDCl_3_, 300 MHz): 5.24 (s, 2H), 5.28 (s, 2H), 7.05 (d, *J* = 8.1 Hz, 1H), 7.34–7.52 (m, *J* = 12.0 Hz, 12 Hz), 9.84 (s, 1H); ^13^C-NMR (CDCl_3_, 75 MHz): 70.4, 70.5, 112.0, 112.7, 126.2, 126.6, 126.8, 127.5, 127.6, 128.1, 128.2, 129.9, 135.8, 136.1, 148.8, 153.9, 190.3 ppm; HR-MS (ESI) calculated for C_21_H_19_O_3_ [M + H] 319.1334, found 319.1330.

Preparation of *3,4-bis(benzyloxy)phenol*
**8**: To a solution of compound **7** (8.60 g, 27.0 mmol) in DCM (135 mL) was added *m*-chloroperbenzoic acid (7.00 g, 40.5 mmol), and then the mixture was stirred at room temperature for 15 h. The reaction was quenched with saturated aqueous Na_2_S_2_O_3_ solution (25 mL) and saturated aqueous Na_2_CO_3_ solution (125 mL). The mixture was firstly extracted with DCM (125 mL) and then successively washed with saturated aqueous Na_2_CO_3_ solution (125 mL × 2) and brine (75 mL). After drying over Na_2_SO_4_, the solvent was removed under vacuum. Next, the residue was firstly dissolved in MeOH (135 mL) and then treated with K_2_CO_3_ (4.10 g, 29.7 mmol). The mixture was stirred at room temperature for 30 min. After the reaction was completed, the solvent was removed under vacuum, and water (100 mL) was added. HCl solution (3 M) was added to adjust the pH to **3**. The mixture was extracted with ethyl acetate (250 mL × 2). The combined organic layer was washed with brine (200 mL) and dried over Na_2_SO_4_. Finally, the extracts were filtered and concentrated to afford the intermediate **8** without any purification for the next step. White solid; 7.19 g, 87% yield; m.p. 105–106 °C; R_f_: 0.25 (EA:hexane = 1:4); IR (cm^−1^): 3031, 2926, 1608, 1508, 1453, 1435, 1281, 1269, 1245, 1165, 1003; ^1^H-NMR (CDCl_3_, 300 MHz): 5.09 (s, 2H), 5.11 (s, 2H), 6.30–6.33 (m, 1H), 6.51 (s, 1H), 6.79–6.83 (m, 1H), 7.28–7.44 (m, 10H); ^13^C-NMR (CDCl_3_, 75 MHz): 70.6, 72.4, 102.9, 106.5, 117.2, 126.8, 127.2, 127.3, 127.4, 127.9, 128.0, 136.5, 137.1, 142.3, 149.8, 150.4 ppm; HR-MS (ESI) calculated for C_20_H_19_O_3_ [M + H] 307.1334, found 307.1336.

Preparation of *1,2-dibenzyloxy-4-(p-methoxybenzyl)oxybenzene*
**9**: To a solution of compound **8** (6.64 g, 21.7 mmol) in DMF (20 mL) was added NaH (60%, dispersion in paraffin liquid) (1.05 g, 26.1 mmol) at 0 °C. The mixture was stirred for 30 min, and PMBCl (3.22 mL, 23.9 mmol) was added. The mixture was continually stirred at room temperature. After the reaction was finished according to TLC monitoring, the saturated NH_4_Cl solution (100 mL) was added, and the mixture was extracted with ethyl acetate (100 mL × 2). The organic layer was then washed with water (100 mL) and brine (50 mL) and dried over Na_2_SO_4_. The filtrate was concentrated, and the residue was stirred in hexane (75 mL) overnight. The mixture was filtered to provide the compound **9** without any purification for the next step. White solid; 8.32 g, 90% yield; m.p. 80–81 °C; R_f_: 0.62 (EA:hexane = 1:4); IR (cm^−1^): 3062, 3035, 2914, 2867, 1610, 1589, 1515, 1468, 1418, 1392, 1380, 1272, 1228, 1171, 1116, 1004; ^1^H-NMR (CDCl_3_, 300 MHz): 3.84 (s, 3H), 4.92 (s, 2H), 5.12 (s, 2H), 5.15 (s, 2H), 6.47–6.51 (m, 1H), 6.68 (d, *J* = 8.7 Hz, 1H), 6.88 (d, *J* = 8.7 Hz, 1H), 6.93 (s, 1H), 6.95 (s, 1H), 7.33–7.46 (m, 12H); ^13^C-NMR (CDCl_3_, 75 MHz): 54.8, 69.8, 70.6, 72.2, 103.4, 105.2, 113.5, 116.6, 126.9, 127.1, 127.2, 127.3, 127.9, 128.0, 128.6, 128.8, 136.6, 137.2, 142.7, 149.7, 153.7, 159.0 ppm; HR-MS (ESI) calculated for C_28_H_27_O_4_ [M + H] 427.1909, found 427.1909. 

Preparation of *1,2-dibenzyloxy-4-bromo-5-(p-methoxybenzyl)oxybenzene*
**1**: To a solution of compound **8** (7.32 g, 17.2 mmol) in DMF (185 mL) was added the solution of NBS (3.21 g, 18.0 mmol) in DMF (35 mL) dropwise at 0 °C. After stirring for 30 min, the mixture was warmed to room temperature and stirred for additional 4 h. When the reaction was completed, ice water (500 mL) was added and a white solid appeared. After the suspension was filtered, the residue was washed with ice water, and dried under reduced pressure to give the intermediate **1**. White solid; 7.54 g, 87% yield; m.p. 103–104 °C; R_f_: 0.62 (EA:hexane = 1:4); IR (cm^−1^): 3058, 3012, 2910, 1589, 1520, 1420, 1395, 1375, 1225, 1180, 1123, 1001; ^1^H-NMR (CDCl_3_, 300 MHz): 3.84 (s, 3H), 4.96 (s, 2H), 5.09 (s, 2H), 5.10 (s, 2H), 6.63 (s, 1H), 6.92 (d, *J* = 8.7 Hz, 2H), 7.17 (s, 1H), 7.33–7.46 (m, 12H); ^13^C-NMR (CDCl_3_, 75 MHz): 54.8, 71.4, 71.5, 72.1, 103.0, 104.4, 113.5, 120.1, 126.9, 127.1, 127.4, 127.5, 128.0, 128.1, 128.2, 128.5, 136.3, 136.5, 143.7, 148.5, 149.5, 158.9 ppm; HR-MS (ESI) calculated for C_28_H_25_BrKO_4_ [M + K] 543.0573, found 543.0559.

Preparation of *5,7-dihydroxy-2-chromenone*
**11**: A stirred mixture of phloroglucinol dihydrate (18.90 g, 150.0 mmol), ethyl propiolate (17.7 mL, 180.0 mmol), and ZnCl_2_ (1.00 g, 7.5 mmol) was heated at 100 °C for 2 h. A flocculent precipitate formed during the reaction. After the mixture was filtered, the solid was recrystallized from water to afford the intermediate **11**. Yellow solid; 21.36 g, 80% yield; m.p. 263–264 °C; R_f_: 0.50 (EA:hexane = 3:1); IR (cm^−1^): 3421, 3054, 1686, 1615, 1574, 1474, 1365, 1302, 1244, 1156, 1071; ^1^H-NMR (DMSO-*d_6_*_,_ 300 MHz): 5.94 (d, *J* = 9.6 Hz, 1H), 6.09 (m, 1H), 6.17 (d, *J* = 2.1 Hz, 1H), 7.86 (dd, *J* = 5.7, 9.6 Hz, 1H), 10.28 (s, 1H), 10.56 (s, 1H); ^13^C-NMR (DMSO*-d_6_*, 75 MHz): 94.0, 98.2, 101.6, 108.6, 139.5, 155.9, 156.4, 160.7, 162.0; HR-MS (ESI) calculated for C_9_H_7_O_4_ [M+H] 179.0344, found 179.0330.

Preparation of *5,7-diacetoxyl-2-chromenone*
**12**: A mixture of **11** (5.00 g, 28.0 mmol) and K_2_CO_3_ (19.40 g, 140.0 mmol) in dry acetone was heated to reflux (oil bath, 90 °C) for 1 h. Acetyl chloride (8.0 mL, 112.3 mmol) was added dropwise with syringe. The reaction suspension was stirred for another 1.5 h at 90 °C. The resulting solution was cooled down and filtrated out of the excess K_2_CO_3_. The solvent was concentrated in vacuo, and the residue was recrystallized by MeOH to give compound **12**. White solid; 6.24 g, 85% yield; m.p. 129–130 °C; R_f_: 0.28 (EA:hexane = 1:2); IR (cm^−1^): 3075, 1767, 1745, 1630, 1575, 1433, 1365, 1309, 1242, 1188, 1131, 1066, 1027; ^1^H-NMR (DMSO*-d_6_*, 300 MHz): 2.31 (s, 3H), 2.40 (s, 3H), 6.51 (d, *J* = 9.6 Hz, 1H), 7.11 (d, *J* = 2.1 Hz, 1H), 7.24 (dd, *J* = 0.6, 2.1 Hz, 1H), 8.07 (dd, *J* = 0.6, 9.6 Hz, 1H); ^13^C-NMR (DMSO*-d_6_*, 75 MHz): 20.6, 20.8, 108.0, 110.6, 112.9, 115.9, 137.9, 147.4, 152.6, 154.3, 159.2, 168.5, 168.8 ppm; HR-MS (ESI) calculated for C_13_H_11_O_6_ [M + H] 263.0556, found 263.0538.

Preparation of *3-bromo-5,7-diacetoxyl-2-chromenone*
**13**: To a solution of **12** (2.00 g, 7.6 mmol) in DCM (20 mL) was added Br_2_ (1.95 mL, 38.0 mmol) at 0 °C. The mixture was stirred at 10 °C for 8 h. The reaction was quenched with a 2 M NaOH aqueous solution (25 mL). The brown solution was extracted with DCM (50 mL). The organic layer was washed with water (25 mL × 2) and brine (25 mL) and then dried over Na_2_SO_4_. The filtrate was concentrated in vacuo to afford the dibromochromanone. To the above dibromide in DCM (25 mL) was added Et_3_N (1.33 g, 13.2 mmol) at room temperature, which was then stirred for 1 h. The resulting solution was removed from the solvent in vacuo, and the residue was recrystallized by MeOH to give the desired product **13**. White solid; 1.94 g, 75% yield; m.p. 130–131 °C; R_f_: 0.51(EA:hexane = 1:2); IR (cm^−1^): 3065, 1765, 1743, 1619, 1430, 1291, 1239, 1192, 1132, 1067, 1022; ^1^H-NMR (CDCl_3_, 300 MHz): 2.34 (s, 3H), 2.44 (s, 3H), 7.04–7.06 (m, 2H), 8.11 (s, 1H); ^13^C-NMR (CDCl_3_, 75 MHz): 20.4, 20.6, 107.3, 110.4, 111.1, 112.2, 137.6, 145.9, 152.4, 153.4, 167.5, 167.6 ppm; HR-MS (ESI) calculated for C_13_H_9_BrNaO_6_ [M + Na] 362.9480, found 362.9476.

Preparation of *3-bromo-5-benzyloxy-7-acetoxyl-2-chromenone*
**3**: To a solution of **13** (1.53 g, 4.5 mmol) and K_2_CO_3_ (1.88 g, 13.5 mmol) in acetone (30 mL) was added BnBr (0.81 mL, 6.8 mmol), and then the mixture was stirred at 56 °C overnight. The suspension was concentrated in vacuo, and water (75 mL) was added. The resulting solution was extracted with ethyl acetate (100 mL × 2). The combined organic layer was washed with brine (100 mL) and dried over Na_2_SO_4_. After the mixture was filtered, the filtrate was removed from the solvent and the residue was recrystallized by EtOH to give the desired product **3**. White solid; 1.13 g, 65% yield; m.p. 176–177 °C; R_f_: 0.60 (EA:hexane = 1:2); IR (cm^−1^): 3091, 2924, 2853, 1737, 1615, 1499, 1459, 1430, 1351, 1296, 1234, 1208, 1131, 1028; ^1^H-NMR (DMSO*-d_6_*, 300 MHz): 2.31 (s, 3H), 5.25 (s, 2H), 6.93 (d, *J* = 1.3 Hz, 1H), 6.98 (d, *J* = 1.6 Hz, 1H), 7.36–7.45 (m, 3H), 7.51–7.54 (m, 2H), 8.39 (s, 1H); ^13^C-NMR (DMSO*-d_6_*, 75 MHz): 20.8, 70.6, 102.8, 102.9, 107.7, 108.7, 127.8, 128.2, 128.5, 135.8, 138.9, 153.9, 154.0, 154.6, 156.3, 168.5 ppm; HR-MS (ESI) calculated for C_18_H_14_BrO_5_ [M + H] 389.0025, found 389.0022.

Preparation of *5-(benzyloxy)-3-(4,5-bis(benzyloxy)-2-((4-methoxybenzyl)oxy)phenyl)-7-hydroxy-2-chromenone*
**16**: To a solution of **1** (6.06 g, 12.0 mmol) and Pd(PPh_3_)_2_Cl_2_ (420 mg, 0.6 mmol, 5 mol%) in THF (120 mL) was added TEA (14.4 mL, 103.0 mmol, 8.6 equivalents) and pinacoborane (13.0 mL, 90.0 mmol, 7.5 equivalents) under Ar atmosphere. The mixture was heated overnight at 80 °C. After being cooled, **3** (3.18 g, 8.0 mmol), Pd(dppf)Cl_2_ (293 mg, 0.4 mmol, 5 mol.%), and a solution of Na_2_CO_3_ (8.90 g, 84.0 mmol) in H_2_O (30 mL) was added under Ar atmosphere. The mixture was stirred at 90 °C for another 12 h, and a 3 M HCl solution was added to adjust the pH to 3. The mixture was filtered with diatomaceous earth, and the filtrate was extracted with ethyl acetate (200 mL × 3). The combined organic layer was washed with brine (200 mL) and then dried over Na_2_SO_4_. After drying and concentrating, the residue was purified by column chromatography with ethyl acetate/petroleum ether (1:3–1:1) to afford the desired product **16**. Yellow solid; 3.98 g, 72% yield; m.p. 166–167 °C; R_f_: 0.32 (EA:hexane = 1:2); IR (cm^−1^): 3425, 3012, 2988, 1725, 1613, 1520, 1443, 1400, 1387, 1312, 1214, 1189, 1102, 1043, 1004; ^1^H-NMR (DMSO*-d_6_*, 300 MHz): 3.69 (s, 3H), 4.95 (s, 2H), 5.03 (s, 2H), 5.19 (s, 2H), 5.20 (s, 2H), 6.37 (s, 1H), 6.46 (s, 1H), 6.77 (s, 1H), 6.80 (s, 1H), 6.98 (s, 1H), 7.09 (s, 1H), 7.20–7.46 (m, 17H), 7.88 (s, 1H); ^13^C-NMR (DMSO*-d_6_*, 75 MHz): 55.0, 70.0, 70.4, 71.4, 93.7, 94.8, 96.6, 98.0, 102.2, 102.5, 113.6, 116.9, 118.5, 127.5, 127.6, 127.7, 127.8, 128.0, 128.2, 128.4, 128.5, 128.9, 129.1, 136.3, 136.6, 137.1, 137.4, 141.8, 149.2, 150.8, 155.5, 155.9, 158.8, 159.8, 161.8 ppm; HR-MS (ESI) calculated for C_44_H_37_O_8_ [M + H] 693.2488, found 693.2484.

Preparation of *5-(benzyloxy)-3-(4,5-bis(benzyloxy)-2-((4-methoxybenzyl)oxy)phenyl)-7-methoxy-2-chromenone*
**4**: To a solution of **16** (1.38 g, 2.0 mmol) and K_2_CO_3_ (0.42 g, 3.0 mmol) in DMF (10 mL) was added MeI (250 uL, 4.0 mmol) slowly, and then the mixture was stirred at the room temperature. After TLC showed the reaction was completed, the mixture was diluted with ethyl acetate (50 mL), and the organic layer was washed with H_2_O (40 mL × 2) and brine (25 mL). After drying over Na_2_SO_4_, the filtrate was concentrated, and the residue was purified by column chromatography with ethyl acetate/petroleum ether (1:3–1:2) to afford the desired product **4**. Yellow solid; 1.20 g, 85% yield; m.p. 130–132 °C; R_f_: 0.46 (EA:hexane = 1:2); IR (cm^−1^): 3031, 3005, 2927, 2835, 1719, 1610, 1513, 1445, 1413, 1385, 1300, 1250, 1197, 1155, 1109, 1082, 1015; ^1^H-NMR (DMSO*-d_6_*, 300 MHz): 3.69 (s, 3H), 3.85 (s, 3H), 4.95 (s, 2H), 5.04 (s, 2H), 5.20 (s, 2H), 5.25 (s, 2H), 6.63 (d, *J* = 0.9 Hz, 2H), 6.76 (s, 1H), 6.79 (s, 1H), 7.00 (s, 1H), 7.11 (s, 1H), 7.20 (s, 1H), 7.23 (s, 1H), 7.30–7.44 (m, 11H), 7.46–7.49 (m, 4H), 7.90 (s, 1H); ^13^C-NMR (DMSO*-d_6_*, 75 MHz): 55.0, 56.0, 70.2, 70.3, 70.4, 71.4, 93.0, 96.3, 102.2, 103.6, 113.6, 116.7, 118.4, 119.6, 127.6, 127.7, 127.8, 128.0, 128.2, 128.4, 128.5, 128.8, 129.1, 136.2, 136.4, 137.1, 137.4, 141.8, 149.4, 150.9, 155.5, 155.6, 158.8, 159.6, 163.0; HR-MS (ESI) calculated for C_45_H_39_O_8_ [M + H] 707.2645, found 707.2648.

Preparation of *1,8,9-tris(benzyloxy)-3-methoxy-6H-benzofuro[3,2-c]chromen-6-one*
**17**: To a solution of **4** (0.71 g, 1.0 mmol) in toluene (20 mL) was added DDQ (0.45 g, 2.0 mmol), and then the mixture was stirred under reflux for 24 h. After the reaction was finished, the mixture was concentrated and then purified by column chromatography (DCM) to give the product **17**. White solid; 327 mg, 56% yield; m.p. 212–213 °C; R_f_: 0.40 (EA:hexane = 1:2); IR (cm^−1^): 3031, 2930, 2850, 1736, 1604, 1450, 1401, 1352, 1301, 1273, 1197, 1161, 1137, 1081, 1049; ^1^H-NMR (CDCl_3_, 300 MHz): 3.88 (s, 3H), 5.25 (s, 2H), 5.27 (s, 2H), 5.31 (s, 2H), 6.48 (d, *J* = 2.1 Hz, 1H), 6.63 (d, *J* = 2.1 Hz, 1H), 7.20 (s, 1H), 7.37–7.63 (m, 15H), 7.68 (s, 1H); ^13^C-NMR (CDCl_3_, 75 MHz): 55.8, 70.8, 71.8, 72.0, 94.1, 97.1, 99.4, 105.5, 116.2, 126.8, 127.3, 127.5, 127.9, 128.0, 128.1, 128.5, 128.6, 128.7, 136.2, 136.8, 137.0, 148.0, 148.5, 150.1, 155.4, 155.9, 158.5, 159.7, 162.8; HR-MS (ESI) calculated for C_37_H_28_NaO_7_ [M + Na] 607.1733, found 607.1718.

Preparation of WEL **5**: To a solution of **17** (0.58 g, 1.0 mmol) in DCM (15 mL) was added BCl_3_ (3 mL, 1 M) slowly at 0 °C under Ar atmosphere. After the reaction was finished, the mixture was concentrated and then purified by column chromatography (MeOH:DCM = 1:20) to give the product **5**. Gray solid; 254 mg, 81% yield; m.p. >300 °C; R_f_: 0.48 (MeOH:DCM = 1:10); IR (cm^−1^): 3397, 2955, 2923, 2853, 1710, 1670, 1612, 1447, 1324, 1284, 1206, 1151, 1070, 1047; ^1^H-NMR (DMSO*-d_6_*, 300 MHz): 3.82 (s, 3H), 6.45 (d, *J* = 1.8 Hz, 1H), 6.62 (d, *J* = 1.9 Hz, 1H), 7.16 (s, 1H), 7.24 (s, 1H); ^13^C-NMR (DMSO*-d_6_*, 75 MHz): 55.7, 93.2, 96.7, 98.1, 98.8, 101.7, 104.5, 113.7, 144.3, 145.4, 148.8, 154.8, 155.2, 157.7, 158.9, 162.2; HR-MS (ESI) calculated for C_16_H_11_O_7_ [M + H] 315.0505, found 315.0504.

Characterization data of WEL reported by Yang’s group were as follows: ^1^H-NMR (DMSO*-d_6_*, 300 MHz): 3.77 (s, 3H), 6.40 (d, *J* = 2.1 Hz, 1H), 6.55 (d, *J* = 1.8 Hz, 1H), 7.15 (s, 1H), 7.23 (s, 1H); ^13^C-NMR (DMSO*-d_6_*, 75 MHz): 55.7, 93.2, 96.7, 98.1, 98.9, 101.7, 104.6, 113.8, 144.3, 145.4, 148.9, 154.8, 155.3, 157.8, 158.9, 162.2.

All above chemicals were commercially available and used without further purification. Analytical thin-layer chromatography was performed on glass plates precoated with silica gel impregnated with a fluorescent indicator (254 nm). The plates were visualized by exposure to ultraviolet light. ^1^H-NMR spectra were recorded on a Bruker DRX (300 MHz), and ^13^C-NMR spectra were recorded on a Bruker DRX (75 MHz). HR-MS spectra were taken on an Agilent Technologies Accurate-Mass Q-TQF LC/MS 6520.

### 3.3. Biological Assays

The tyrosinase inhibition activity of the compounds was measured using l-DOPA as a substrate according to a modified method of previous work [4,10,11]. WEL was firstly dissolved in DMSO at a concentration of 1.0 mM, and the final concentration of DMSO in the reaction mixture was 3%. In the investigation, the total volume of the reaction system was 300 µL; l-DOPA was used as the substrate for the determination of diphenolase activity. Briefly, l-DOPA (100 µL, 0.5 mM), phosphate buffer (180 µL, pH 6.8, 50 mM), and different concentrations of inhibitors (10 µL in DMSO) were mixed and incubated at 30 °C. Then, an aqueous solution of mushroom tyrosinase (10 µL, 1000 U/mL) was added to the above solution and mixed quickly. Absorption and kinetic measurements were carried out on a spectrophotometer from Thermo Fisher at 475 nm. Control experiments, containing the same amount of DMSO without inhibitor, were routinely carried out. All measurements were performed in triplicate. The enzyme activity was calculated according to Lambert–Beer’s law (ε = 3700 L∙mol^−1^∙cm^−1^). The extent of inhibition by the inhibitor was expressed as the concentration required to reduce tyrosinase activity to 50% (IC_50_). The percentage inhibition of tyrosinase activity was calculated as follows:
% inhibition = [1 − (X − A_1_)/(A_2_ − A_1_)] × 100%,
where X is the absorbance at 475 nm of the inhibitor, A_1_ is the absorbance at 475 nm of the solution without tyrosinase, and A_2_ is the absorbance at 475 nm of the bank (without inhibitor).

## 4. Conclusions

In summary, we developed an efficient synthetic route to synthesize wedelolactone, wherein Pd(II)-catalyzed boronation/coupling reactions and DDQ-involved oxidative deprotection/annulation are key reactions. The obtained WEL was determined as an efficient tyrosinase inhibitor which could inhibit the enzyme in a reversible, competitive manner. Computational docking simulations showed that WEL could bind with tyrosinase because its hydroxyl group on the scaffold bound residue Asn260 located in the active site. The study suggested that WEL may be efficient in restricting enzymatic browning reactions, which could be widely used in multiple fields.

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
