# Peer review of "Palladium(II)-Catalyzed Efficient Synthesis of Wedelolactone and Evaluation as Potential Tyrosinase Inhibitor"

_molecules, 2019, doi:10.3390/molecules24224130_

Round 1

Reviewer 1 Report

In the Manuscript entitled ‘Palladium (II)-catalyzed efficient synthesis of Wedelolactone and evaluation as potential tyrosinase inhibitor’ by Ji and co workers have developed a new way to synthesize wedelolactone which is an efficient tyrosinase inhibitor.
This manuscript is suitable to be published in Molecules with minor corrections.

Line 16: Before we studying should be changed to before we studied.

Line 40: Have been proved to should be changed to Have proven to be.

Line 52: Authors claim that the total yield is relatively low for the method developed by Yang group. Its better if they state the longest linear sequence and the overall yield for their compared to the other group.

Line 77: Benzaldehyde spelling should be corrected

Line 79: Firstly protected with should be changed to first protected with

Line 87: Authors have claimed that the Acetoxyl-coumarin 12 was obtained in quantitative yield, I hope authors understand the meaning of quantitative yield, 85% is certainly good but not quantitative.

Line 97: byproducts spelling should be corrected

Line 140: space in the spelling Versus should be corrected

Line 154: WasAsn260 should have space

For supporting information Rf, and IR should be reported for all compounds. Melting point should be specified instead of just writing the numbers.

Author Response

Reviewer 1:

Suggestions: in the Manuscript entitled ‘Palladium (II)-catalyzed efficient synthesis of Wedelolactone and evaluation as potential tyrosinase inhibitor’ by Ji and co workers have developed a new way to synthesize wedelolactone which is an efficient tyrosinase inhibitor.

This manuscript is suitable to be published in Molecules with minor corrections.

Line 16: Before we studying should be changed to before we studied.

Line 40: Have been proved to should be changed to Have proven to be.

Response: Sorry for our mistakes and the sentences have been correct as the reviewer’s suggestions.

Line 52: Authors claim that the total yield is relatively low for the method developed by Yang group. Its better if they state the longest linear sequence and the overall yield for their compared to the other group.

Response: Thanks for the reviewer’s suggestions. The overall yield of this route was listed in the revised manuscript.

“The route is in low 15% overall yield with longest linear sequence (total 12 steps) and it’s hardly applied to give an access to a variety of WEL analogues for structure transformations.”

Line 77: Benzaldehyde spelling should be corrected

Line 79: Firstly protected with should be changed to first protected with

Response: Sorry for our mistakes and the sentences have been correct as the reviewer’s suggestions.

Line 87: Authors have claimed that the Acetoxyl-coumarin 12 was obtained in quantitative yield, I hope authors understand the meaning of quantitative yield, 85% is certainly good but not quantitative.

Response: Sorry for our imprecise description and the “quantitative yield” has been changed to “in a good yield”

Line 97: byproducts spelling should be corrected

Line 140: space in the spelling Versus should be corrected

Line 154: WasAsn260 should have space

Response: We have changed as the reviewer’s suggestions in the revised manuscript.

Reviewer 2 Report

The manuscript by Huidan Huang and co-workers presented an original studies deals with the total synthesis of a natural product Wedelolactone and with the study of these compound as potential tyrosinase inhibitor in silico and in vitro. The main novelty of this research is the development of one-pot boronation/Suzuki-Miyaura reaction.

Of interest is also the proposed methodology of DDQ-involved deprotection and oxidative annulation consequents. All the obtained results are clarified and well presented. The obtained structures are well characterized by spectral data. 

Before publication in Molecules I strongly suggest additions and correlations.

(1) The reference [21] have not deal with DDQ-mediated remove of PMB group. It is a mistake. An another reference must be included.

(2) All synthesized compounds: 1,3,4,7,8,9, 11,12,13,16,17 – are not call intermediates; the name of the individual compounds must be added. Moreover, for known compounds the subsequent reference must be added. 

Author Response

Reviewer 2:

The manuscript by Huidan Huang and co-workers presented an original studies deals with the total synthesis of a natural product Wedelolactone and with the study of these compound as potential tyrosinase inhibitor in silico and in vitro. The main novelty of this research is the development of one-pot boronation/Suzuki-Miyaura reaction.

Of interest is also the proposed methodology of DDQ-involved deprotection and oxidative annulation consequents. All the obtained results are clarified and well presented. The obtained structures are well characterized by spectral data. Before publication in Molecules I strongly suggest additions and correlations.

Responses: Thanks for the reviewer’s positive comments on our manuscript.

(1) The reference [21] have not deal with DDQ-mediated remove of PMB group. It is a mistake. An another reference must be included.

Responses: Sorry for our mistakes and the reference [21] has been changed to “Green, R.A.; Jolley, K.E.; Al-Hadedi, A.A.M.; Pletcher, D.; Harrowven, D.C.; Frutos, O.D.; Mateos, C.; Klauber, D.J.; Rincon, J.A.; Brown, R.C.D. Electrochemical deprotection of para-methoxybenzyl ethers in a flow electrolysis cell. Org. Lett. 2017, 19, 2050-2053.”

(2) All synthesized compounds: 1,3,4,7,8,9, 11,12,13,16,17 – are not call intermediates; the name of the individual compounds must be added. Moreover, for known compounds the subsequent reference must be added.

Response: Thanks for the reviewer’s suggestions and all intermediates have been added the names. The known compounds have been added subsequent reference

Reviewer 3 Report

The paper by Q. Chen and F. Ji et al reports the total synthesis of wedelolactone, a natural product with tyrosinase inhibitory activity. The strategy relies on a twelve steps sequence of chemical transformations, based on a disconnection that requires to prepare an arylboronate and a brominated coumarin. The major drawback of this paper is that this approach has already been described in the literature, using an aryltin reagent instead of the corresponding arylboronate. Even though the results are clearly presented, with nevertheless many English errors, originality is low and improvements from already published work do not justify a publication in Molecules. These results could be published in another journal.

Author Response

Reviewer 3:

The paper by Q. Chen and F. Ji et al reports the total synthesis of wedelolactone, a natural product with tyrosinase inhibitory activity. The strategy relies on a twelve steps sequence of chemical transformations, based on a disconnection that requires to prepare an arylboronate and a brominated coumarin. The major drawback of this paper is that this approach has already been described in the literature, using an aryltin reagent instead of the corresponding arylboronate. Even though the results are clearly presented, with nevertheless many English errors, originality is low and improvements from already published work do not justify a publication in Molecules. These results could be published in another journal.

Response: Thanks for the reviewer’s comments on our manuscript. In the revised manuscript, we have corrected our English errors. The manuscript focused on an improved synthetic route to synthesize wedelolactone (WEL). In addition, we firstly discovered that the natural product WEL was an efficient tyrosinase inhibitor which could be efficient in restricting enzymatic browning reactions. Thus, we are confident that our revised manuscript fits the criteria of Molecules.

Reviewer 4 Report

This manuscript reports an alternative synthetic route to Wedelolactone and evaluation of this molecule as a potential tyrosinase inhibitor. The two known routes to this molecule have limitations, so it’s reasonable to continue to search for a better approach. However, I am not convinced if this paper provides one. In order to convince me, the authors have to provide an evaluation of the overall synthetic efficiency for the three syntheses: the number of steps, the total yield and, perhaps, the average yield per step.  Once this is done, the paper should be publishable in Molecules since it provides a useful combination of synthetic work and biological studies.

Author Response

Reviewer 4

This manuscript reports an alternative synthetic route to Wedelolactone and evaluation of this molecule as a potential tyrosinase inhibitor. The two known routes to this molecule have limitations, so it’s reasonable to continue to search for a better approach. However, I am not convinced if this paper provides one. In order to convince me, the authors have to provide an evaluation of the overall synthetic efficiency for the three syntheses: the number of steps, the total yield and, perhaps, the average yield per step.  Once this is done, the paper should be publishable in Molecules since it provides a useful combination of synthetic work and biological studies.

Response: Thanks for the reviewer’s positive comments on our manuscript. As for the reviewer’s concern on our synthetic routs, we would express as follows:

       Actually, the two routes listed in Figure 1 have been commonly recognized by industry. However, both of these two methods had several disadvantages. The first method (reported by Yang group) has been achieved, in which the longest linear sequence is nine steps, in about 15% overall yield. The route in the first step involved the synthesis of the crucial intermediate phenyl acetylene, which was hard to isolate and could be prepared in only 35% yield. The disadvantage limited the application of this route in the synthesis of WEL in a large scale and was detrimental to give an access to a variety of WEL analogues for structure transformations.

The second method (reported by by Lee’s et al) has been covered, in which the longest linear sequence is nine steps, in about 50% overall yield. However, the route was utilized by our group to prepare WEL at first and the yield (13% yield) was found to be much lower than the reported yield. Moreover, the method employed toxic organotin reagents, which limited the industrial production and increased the operation complexity. Due that the present methods are imperfect and unsatisfactory for further investigation of WEL as efficient tyrosinase inhibitors, the development of a facile, versatile, and mild approach is urgently needed. The efficient synthetic route to synthesize wedelolactone wherein Pd(II)-catalyzed borontion/coupling reactions and DDQ-involved oxidative deprotection/annulation are key reactions has been reported in this paper. The route involved eight steps and the desired product could be obtained in about 17% yield. Meanwhile, this method could be easily used to synthesize the WEL analogues for structure modifications. The work about the structure modifications is in progress in our group.

Special thanks to you for your good and valuable comments.